# Automated Machine Learning System for Defect Detection on Cylindrical Metal Surfaces

**DOI:** 10.3390/s22249783

**Published:** 2022-12-13

**Authors:** Yi-Cheng Huang, Kuo-Chun Hung, Jun-Chang Lin

**Affiliations:** 1Department of Mechanical Engineering, National Chung Hsing University, Taichung City 40227, Taiwan; 2Department of Mechatronics Engineering, National Changhua University of Education, Changhua City 50007, Taiwan

**Keywords:** automated machine learning (AutoML), convolutional neural network (CNN), metal surface defect

## Abstract

Metal workpieces are indispensable in the manufacturing industry. Surface defects affect the appearance and efficiency of a workpiece and reduce the safety of manufactured products. Therefore, products must be inspected for surface defects, such as scratches, dirt, and chips. The traditional manual inspection method is time-consuming and labor-intensive, and human error is unavoidable when thousands of products require inspection. Therefore, an automated optical inspection method is often adopted. Traditional automated optical inspection algorithms are insufficient in the detection of defects on metal surfaces, but a convolutional neural network (CNN) may aid in the inspection. However, considerable time is required to select the optimal hyperparameters for a CNN through training and testing. First, we compared the ability of three CNNs, namely VGG-16, ResNet-50, and MobileNet v1, to detect defects on metal surfaces. These models were hypothetically implemented for transfer learning (TL). However, in deploying TL, the phenomenon of apparent convergence in prediction accuracy, followed by divergence in validation accuracy, may create a problem when the image pattern is not known in advance. Second, our developed automated machine-learning (AutoML) model was trained through a random search with the core layers of the network architecture of the three TL models. We developed a retraining criterion for scenarios in which the model exhibited poor training results such that a new neural network architecture and new hyperparameters could be selected for retraining when the defect accuracy criterion in the first TL was not met. Third, we used AutoKeras to execute AutoML and identify a model suitable for a metal-surface-defect dataset. The performance of TL, AutoKeras, and our designed AutoML model was compared. The results of this study were obtained using a small number of metal defect samples. Based on TL, the detection accuracy of VGG-16, ResNet-50, and MobileNet v1 was 91%, 59.00%, and 50%, respectively. Moreover, the AutoKeras model exhibited the highest accuracy of 99.83%. The accuracy of the self-designed AutoML model reached 95.50% when using a core layer module, obtained by combining the modules of VGG-16, ResNet-50, and MobileNet v1. The designed AutoML model effectively and accurately recognized defective and low-quality samples despite low training costs. The defect accuracy of the developed model was close to that of the existing AutoKeras model and thus can contribute to the development of new diagnostic technologies for smart manufacturing.

## 1. Introduction

The application of computer-aided design and analysis in certain domains, such as signal processing and simulation, has gradually increased. Manual product inspections require considerable labor, and inaccurate testing results might be obtained because of human error which can affect the quality of manufactured products. Therefore, the use of automated optical inspection has increased. With advancements in hardware and software, deep learning models have been combined with optical inspection systems to relieve the bottleneck of defect detection in manufacturing. Technology used to detect metal surface defects has surpassed the limits of the human eye. Image classification through deep learning can improve the accuracy of image detection [1,2]. In addition, with the advancement of graphics processing units, the computing power of hardware has considerably increased. The You Only Look Once algorithm [3] and deep learning frameworks, such as TensorFlow [4] and PyTorch [5], have been used for defect detection. Synergistic development using a kernel filter, pooling, or activation function in image classification has promoted advances in deep learning technology. Many studies have employed convolutional neural networks (CNNs) to classify images [6,7,8]. CNNs have deep learning structures and can be easily trained [9,10]. Such networks have been used to effectively inspect products and detect defects in images [11].

Theoretically, the number of hidden layers of an NN strongly influences network performance. With more layers, a network can work with and extract more complex feature patterns and therefore achieve superior results. However, the accuracy of a network peaks at a certain number of layers and even decreases thereafter. ResNet [12] uses residual learning to resolve this problem and it contains shortcuts. Therefore, ResNet can suppress the accuracy drop caused by multiple layers in deep networks. When a large kernel is used for feature extraction in convolution operations, numerous parameters are required. MobileNet [13] uses depthwise separable convolution to divide the convolution kernel into single channels. It can convolve each channel without changing the depth of the input features. Moreover, the aforementioned model can produce output feature maps with the same number of channels as the input feature maps. This model can increase or reduce the dimensionality of feature maps to reduce computational complexity and accelerate calculation while maintaining high accuracy.

A deep learning approach was developed for an optical inspection system for surface defects on extruded aluminum [14]. A simple camera records extruded profiles during production, and an NN distinguishes immaculate surfaces from surfaces with various common defects. Metal defects can vary in size, shape, and texture, and the defects detected by an NN can be highly similar. In [15], an automatic segmentation and quantification method using customized deep learning architecture was proposed to detect defects in images of titanium-coated metal surfaces. In [16], a U-Net convolutional network was developed to segment biomedical images through appropriate preprocessing and postprocessing steps; specifically, the network applied a median filter to input images to eliminate impulse noise. Standard benchmarks were used to evaluate the detection and segmentation performance of the developed model, which achieved an accuracy of 93.46%.

In [17], a 26-layer CNN was developed to detect surface defects on the components of roller bearings, and the performance of this network was compared with that of MobileNet, VGG-19 [18], and ResNet-50. VGG-19 achieved a mean average precision (mAP) of 83.86%; however, its processing time was long (i.e., 83.3 ms). MobileNet exhibited the shortest processing speed but the lowest mAP because of the small number of parameters and necessary calculations. The 26-layer CNN achieved a better balance between mAP and processing efficiency than the other three models, with the mAP of this network nearly equal to the highest mAP of ResNet-50. Moreover, the 26-layer CNN required less time for detection than ResNet-50 or VGG-19. In [19], an entropy calculation method was used in a self-designed DarkNet-53 NN model, and the most suitable kernel size was selected for the convolutional layer. The model was highly accurate in recognizing components and required only a short training time.

In [20], two types of residual fully connected NNs (RFCNNs) were developed: RFCN-ResNet and RFCN-DenseNet. The performance of these networks in the classification of 24 types of tumors was compared with that of the XGBoost and AutoKeras automated machine-learning (AutoML) methods. RFCN-ResNet and RFCN-DenseNet featured enhancements in feature propagation and encouragement for the reuse of RFCN architectures, while new RFCN architecture generation achieved accuracies of 95.9% and 95.7%, respectively, outperforming XGBoost and AutoKeras by 4.8% and 4.9%, respectively. In another comparison, RFCN-ResNet and RFCN-DenseNet achieved respective accuracies of 95.9% and 96.5% and outperformed XGBoost and AutoKeras by 6.1% and 5.5%, respectively, indicating that RFCN-ResNet and RFCN-DenseNet considerably outperform XGBoost and AutoKeras in modeling genomic data.

In [21], AutoKeras and a self-designed model were used to analyze water quality. Compared to that of AutoKeras, the accuracy of the developed model was 1.8% and 1% higher in the classification of two-class and multiclass water data, respectively. However, the AutoKeras model exhibited higher efficiency than the developed model and required no manual effort.

The authors of [22] proposed that random trials are more efficient than trials based on a grid search for optimizing hyperparameters. In Gaussian process analysis, different hyperparameters are crucial for different datasets. Thus, a grid search is a poor choice for the configuration of algorithms for new datasets.

In smart manufacturing, quickly adapting to new complex manufacturing processes and designing appropriate and efficient optimization networks have become crucial. In the present study, industrial machine vision and deep learning were combined to construct an AutoML model to detect defects on metal surfaces to reduce costs in smart manufacturing. The proposed model can be used to develop highly adaptable visual inspection techniques to overcome the bottlenecks caused by current image-processing techniques and thereby advance smart manufacturing.

## 2. Introduction to VGGNet, ResNet, MobileNet, and AutoML

### 2.1. VGGNet

VGGNet was developed by the Visual Geometry Group of Oxford University and placed second in the ImageNet Large Scale Visual Recognition Challenge (ILSVRC) in 2014. VGGNet contains more layers than AlexNet did in 2012. The VGG block architecture used by VGGNet contains a repeated 3 × 3 kernel size for the convolutional layers and a 2 × 2 kernel size for the max-pooling layer. Four varieties of VGGNet with different numbers of layers exist: VGG-11; VGG-13; VGG-16; and VGG-19. Among these, VGG-16 and VGG-19 exhibit excellent results. In the present study, the VGG-16 model with few parameters was used for training. VGG-16 contains five VGG blocks as presented in Table 1. Two convolutional layers are used in each of the first two VGG blocks, and three convolutional layers are used in each of the last three VGG blocks. VGG-16 consists of 3 fully connected layers in addition to the 13 convolutional layers.

### 2.2. ResNet

ResNet was developed by Microsoft Research in 2015 and won the ILSVRC that year Beyond a certain point, the accuracy of NNs does not increase with the number of layers. As displayed in Figure 1, the training error of a 56-layer NN is higher than that of a 20-layer NN. This eventual decrease in training accuracy with network depth is the degradation problem of NNs. When the depth of an NN increases, gradients vanishing during backpropagation becomes more likely; thus, certain gradients cannot be transmitted to the next node to update the weights, resulting in a decrease in training accuracy.

To solve the degradation problem, ResNet employs a residual learning structure. As displayed in Figure 2, the input *x* is passed through two branches. In one branch (right), the input *x* is passed across the network layers through a shortcut, and no operation is performed. In the other branch (middle), the output *F*(*x*) is obtained after an operation is performed on *x*. The final output is the sum of the outputs of the two branches, namely *F*(*x*) + *x*. This method can prevent gradient vanishing during convolution operations because the shortcut allows gradients to be passed to the next layer to update the weights.

As displayed in Table 2, the input of ResNet is first passed through a 7 × 7 × 64 convolutional layer and then passed through four residual connection blocks in sequence. The higher the number of layers, the higher the training cost; therefore, a 1 × 1 convolutional layer is placed before a 3 × 3 convolutional layer. The 1 × 1 convolutional layer reduces the number of channels, which can reduce the number of parameters required during training.

### 2.3. MobileNet

Since the introduction of CNNs, the depth of networks has increased. Numerous layers incur a high computational cost, limiting the application of NNs. In 2017, Google developed MobileNet, which is a lightweight NN applied to mobile terminals. The architecture of MobileNet is presented in Table 3. Moreover, depthwise separable convolution is used in place of conventional convolution. The difference between depthwise convolution and general convolution is that depthwise convolution involves splitting the convolution kernel into single-channel forms. Depthwise separable convolution splits kernels to undergo depthwise convolution and pointwise convolution. Even with the same input feature depth, the convolution of each channel enables the generation of output feature maps with the same number of channels as the input feature maps. Pointwise convolution is a 1 × 1 convolution that can increase or reduce the dimensionality of a feature map (Figure 3).

### 2.4. AutoML

The training steps involved in traditional ML are displayed in Figure 4. A training dataset is created, a suitable model is selected for training, and finally, the hyperparameters are adjusted after the training results have been evaluated. This process is repeated with many models and parameters until the most effective ones are identified.

During model selection, aspects such as the dataset size and type, as well as hardware limitations, must be considered. Model evaluation requires an adjustment of the hyperparameters, such as the learning rate and optimizer, as well as the parameters related to the model architecture, such as the number of layers and operation of each layer. In general ML, the aforementioned parameters must be set manually. If the training results are not ideal, transfer learning (TL) can be applied. TL reduces the time required for parameter adjustment. Modeling requires considerable time. Hyperparameter adjustment is the process of searching for combinations, but this process can be automated. AutoML involves using artificial intelligence algorithms to conduct ML automatically.

AutoML enables developers familiar or unfamiliar with NNs, or lacking relevant domain knowledge, to use ML and deep learning techniques. Many tools are available for AutoML. In this study, AutoKeras was used to implement AutoML. AutoKeras is an AutoML system based on the Keras deep learning framework and uses an efficient neural architecture search (ENAS) [23] for automated modeling. AutoKeras employs three common methods to optimize hyperparameters: grid search; random search; and Bayesian search (Figure 5). Grid search is a brute-force method used to check all possible combinations of the range of hyperparameters provided by network designers. For example, if the learning rate is 0.01 or 0.1 and the batch size is 10 or 20, the four possible parameter combinations are produced in sequence for the training method. Random search is similar to grid search, but the combinations of hyperparameters are produced in a random order. Bayesian search involves searching for hyperparameters on the basis of Bayes’ theorem [24], and only parameter combinations that maximize a certain probability function are considered.

In 2016, Google developed the neural architecture search (NAS) with reinforcement learning [25]. The NAS system consists of three main components as displayed in Figure 6. This system includes different types of network layers, including convolutional and fully connected layers. These layers are connected to form network architecture, which generally requires a manual design. In Google’s NAS, various candidate network architectures are tested, and the optimal architecture is selected on the basis of various evaluation indicators. NAS can be used to evaluate chosen strategies based on different models and tests in addition to candidate models.

From target data, NASNet [26] constructs a high-accuracy, high-complexity, and multilayered NN model for image classification. When the number of data is large, the network consumes considerable computation resources. Therefore, it begins searching a small dataset for suitable network layer units and then searches a larger dataset (Figure 7). NASNet judges whether the NN architecture can produce suitable gradient-descent results and modifies the probability of selecting that network architecture according to its judgment. It then selects the optimally performing network model. ENAS is an improved variant of NASNet which allows a parent model to share weights with its submodels; thus, training need not be restarted from scratch for the submodels.

## 3. Architecture of Designed System

### 3.1. Workpiece for Experimental Detection

An experiment was performed to detect defects on a workpiece made of 304 stainless steel parts (Figure 8). A burr generated around a chamfered hole (blue square in Figure 8) during machining with computer numerical control (CNC) was the detection target. Because wires would eventually pass through such a hole, defects must be detected to prevent wire scratching.

The experimental system comprised a 1280 × 1024-pixel camera (Basler acA1280-60gm GigE, CMOS, Ahrensburg, Germany) with a 50 + 15 mm extension ring lens. We used a shadowless ring light to provide illumination from different angles. The surface of the tested workpiece was composed of opaque reflective material. Using a common light source makes the incident angle equal to the reflection angle, thus producing a reflection. Using a ring light source can prevent reflection and highlight defects, thereby effectively solving the reflection problem caused by direct illumination. The optimal light source position and camera inclination angle (α) can be determined through iterative adjustment and experimentation. The angular positioning of the camera relative to the light source in the present study is illustrated in Figure 9.

### 3.2. Dataset

The original image size was 1280 × 550 pixels (Figure 10). The focuses of okay (OK) and not good (NG) images are shown in the green and red boxes, respectively, in Figure 11. After obtaining 300 OK and 300 NG images, we cropped the images to 186 × 189 pixels and used them for training. We defined rough surfaces (e.g., the irregular white parts in the red boxes in Figure 11) as defects.

### 3.3. Experimental Architecture

Three experiments were conducted using Python on Google Colaboratory through the website. In Experiment 1, the results of training different models through TL were evaluated. In each training process, the following hyperparameters were used: the optimizer was Adam; the batch size was 10; the number of epochs was 5; and the learning rate was 0.0001. Datasets for two different training and validation distribution ratios of 5:5 and 8:2 were tested, and the ratio achieving superior results was used for Experiments 2 and 3. Experiment 2 involved the design of this research for an AutoML model. Experiment 3 involved the use of commercial AutoKeras software to import the same dataset for training and a comparison of the model architecture and calculation results with those of our designed AutoML model.

### 3.4. Design of AutoML Model

We extracted feature modules from the three models used for TL in Experiment 1, namely VGG-16, ResNet, and MobileNet as V, R, and M, respectively. In Figure 12, the VGG block, residual connection block of ResNet, and depthwise separable convolution block of MobileNet (i.e., the extracted feature modules) are denoted by V, R, and M, respectively. The designed AutoML model contained a network of the aforementioned blocks. Adam, SGD (stochastic gradient decent), and Adagrad (Adaptive gradient) were selected as the optimizers for this model, and the learning rates were 0.01, 0.001, and 0.0001. In [22], random search was more effective than grid search for the selection of hyperparameters. Therefore, we used random search to select the model architecture, optimizer, and learning rate. If the model accuracy was insufficient, a new architecture and new hyperparameters were used for retraining. In Experiment 1, VGG-16 exhibited an accuracy of 91% when TL was used; thus, 91% was established as the standard for retraining the designed AutoML model. The operation of the designed AutoML model is illustrated in Figure 12.

## 4. Experimental Results and Discussion

The following text describes the training accuracy, validation accuracy, loss charts, and confusion matrices obtained for each model with the number of epochs being 5. The confusion matrix comprises true positives (TPs), true negatives (TNs), false positives (FPs), and false negative (FNs). First of all, in Experiment 1, the TPs were OK images correctly identified as OK. The TNs were NG images correctly identified as NG. The FPs were NG images incorrectly identified as OK. Finally, the FNs were OK images incorrectly identified as NG.

### 4.1. Results of Experiment 1

Figure 13 displays the accuracy and loss of VGG-16, ResNet-50, and MobileNet v1 in training and validation. The training and validation accuracy of VGG-16 approached 1 when the number of epochs was 3 and decreased marginally when the number of epochs was 4. When the number of epochs was 5, the accuracy was higher and lower than when the number of epochs was 4 and 3, respectively. The training loss was approximately 0.6 with 4 epochs. The training and verification accuracy of ResNet-50 approached 1 when the number of epochs was 5 and 4 respectively. The validation accuracy increased to 0.6 when the number of epochs was 3. When the number of epochs was 5, the validation loss was poor at 0.6, which is not favorable. The training accuracy of MobileNet v1 increased from 0.5 at the beginning to 0.8. The validation accuracy initially increased with training accuracy; however, when the number of epochs was 3, the validation accuracy was only approximately 0.4. The validation and training losses of MobileNet v1 were close to 0.8 and above 0.4, respectively, when the number of epochs was 5. Thus, by TL, the phenomenon of apparent convergence in prediction accuracy, followed by divergence in validation accuracy, could happen, especially when the selected network model and content of the image profile is not known prior.

The accuracy, loss, number of parameters, and training time for each tested model are listed in Table 4. In terms of accuracy and loss, the VGG-16 model exhibited optimal performance; however, it required a high number of parameters (i.e., 134,268,738) and a long training time. The MobileNet v1 model required the shortest training time of only 77 s but achieved low accuracy.

Table 5 presents the training results obtained with VGG-16, ResNet-50, and MobileNet v1 under two distribution ratios. The highest values for the 5:5 and 8:2 data distributions were 100 and 20, respectively. The accuracy achieved on the 5:5 dataset was higher than that achieved on the 8:2 dataset; therefore, the 5:5 dataset distribution was used in Experiments 2 and 3. A possible reason for these results is that the defects in the original images comprised with the characteristics of uncomplicated profile as simply as line edges or curves. Overfitting is likely when the traditional 8:2 or 7:3 ratio is used.

### 4.2. Results of Experiment 2

As previously mentioned, the phenomenon of apparent convergence in prediction accuracy, followed by divergence in validation accuracy, which may cause a problem in the selection of an appropriate network model when using TL, sounds reasonable. As presented in Table 5, among the compared models, VGG-16 exhibited the highest accuracy of 0.91 in Experiment 1. Therefore, in Experiment 2, we used this accuracy as the criterion for evaluating the necessity of retraining. Table 6 presents the results obtained with the designed AutoML model during training as well as the hyperparameters, model architecture, training time, accuracy, loss, and confusion matrix for different iterations.

Table 6 lists the model architecture and parameters randomly selected by the designed AutoML model in each of the six training iterations. The RMV (quoted name as the abstracted module adopted from Resnet-50, VGG-16, and MobileNet v1, respectively) model was selected three times, and different optimizers were used for training. However, the training results were poor: the model achieved an accuracy lower than that achieved in Experiment 1. The third training iteration required the highest number of parameters (>2 million) and longest training time (>1 h) but achieved the second highest accuracy (i.e., 90%). In the sixth iteration (Table 7), the MRV model was used with a learning rate of 0.001, Adam was selected as the optimizer, and a considerably lower parameter number was required than those of the first five iterations. The developed AutoML model required 13,242 s (nearly 4 h) to train six models and achieved a final accuracy of 95.5%. The designed AutoML model by MRV is demonstrated in Figure 14.

### 4.3. Results of Experiment 3

AutoKeras has three preset model architectures, namely ResNet-50, EfficientNet B7 [27], and a CNN composed of two convolutional layers. In the present study, AutoKeras selected the CNN model for training. The details of the architecture of the CNN model are presented in Table 8. The dataset input to the CNN model was normalized and then passed through the following layers in sequence: two 3 × 3 convolutional layers; a max-pooling layer; a dropout layer with a dropout rate of 0.25; a flat layer; a dropout layer with a dropout rate of 0.5; and a fully connected layer (named dense), which provided the output. The Adam optimizer and a learning rate of 0.001 were used.

The training results of the AutoKeras model are displayed in Figure 15. This model was trained twice, and no validation dataset was used in the second training iteration. Because AutoKeras merged the training and validation datasets for the final training iteration, Figure 14 displays only a training chart and no validation chart. The trained model weights were added to the test dataset, and a confusion matrix was obtained. The test dataset comprised 100 OK and 100 NG images, and all pictures were identified correctly. Because the AutoKeras model was trained twice, it required a long training time (i.e., 1388 s). The final accuracy and loss of the aforementioned model were 0.9983 and 0.0063, respectively.

## 5. Conclusions

In this study, we detected a burr around the chamfered hole of 304 stainless steel parts produced through CNC machining. To prevent the scratching of the wires that pass through this hole, defects should be detected through imaging; thus, images of the hole were used as the training data in this study. A CNN that can perform TL and AutoML, as well as adopt the AutoKeras model, was designed. Experiments were conducted using these three networks and a training dataset.

The TL model was trained with the following fixed hyperparameters: the Adam optimizer; batch size = 10; number of epochs = 5; and learning rate = 0.0001. This model used VGG-16, ResNet-50, or MobileNet v1. VGG-16 had the highest accuracy among these in the dataset used in this study. The training and validation accuracies of this model were high. Although the training accuracy of ResNet-50 eventually reached 1, its validation accuracy was low. Moreover, large fluctuations in prediction accuracy were observed when the number of epochs was 3. Presumably, larger network layers should cause the result of apparent stability in prediction convergence. However as the number of epochs increase, actually unstable prediction accuracy was observed when the detection defect is not complicated. The training and validation accuracies of MobileNet v1 were considerably lower than those of the other tested models. The designed AutoML model used random search to obtain a combination of modules to construct the optimal model architecture. It then obtained the hyperparameters after training and established a retraining mechanism so that a new architecture could be selected and retrained if the accuracy of the training results was low. The designed AutoML model trained six models and achieved a final training accuracy of 95.5%. The AutoKeras model required a longer training time but constructed the neural architecture search model in a shorter time. The accuracy and two-layer architecture of the convolutional model selected by AutoKeras indicate that the dataset used was simple and did not require a complex model.

The VGG-16, ResNet-50, and MobileNet v1 models all exhibit the advantage of small architecture that prevents gradient vanishing. Based on the spirit of TL, deploying the above model can provide preliminary prediction accuracy in a few trials. Our designed AutoML model composed of a core layer module, obtained by combining the modules of VGG-16, ResNet-50, and MobileNet v1, can effectively improve defect detection and reduce the associated training costs. Our model has considerable advantages in deploying proof-of-concept in defect detection with the selection of a bettering candidate for the CNN. The results of this study can act as a reference for the development of new diagnostic technology for cutting-edge smart manufacturing.

## Figures and Tables

**Figure 1 sensors-22-09783-f001:**
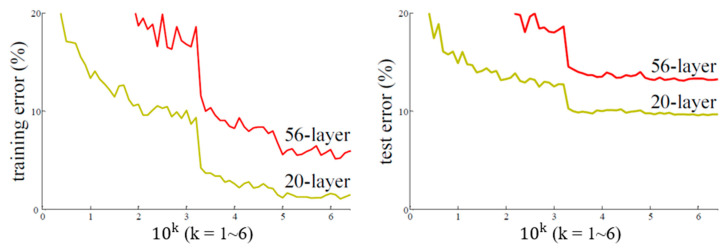
Relationship between number of layers and training error [12].

**Figure 2 sensors-22-09783-f002:**
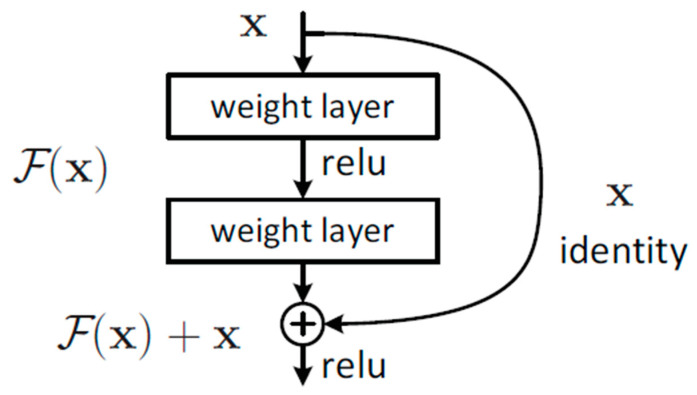
Residual learning structure [12].

**Figure 3 sensors-22-09783-f003:**
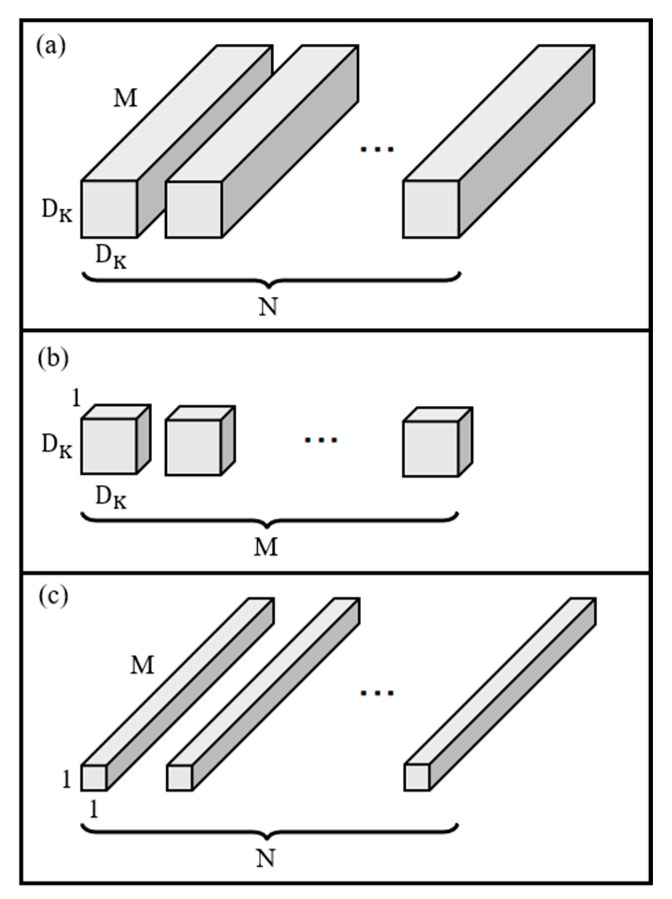
(**a**) General convolution, (**b**) depthwise convolution, and (**c**) pointwise convolution [13].

**Figure 4 sensors-22-09783-f004:**
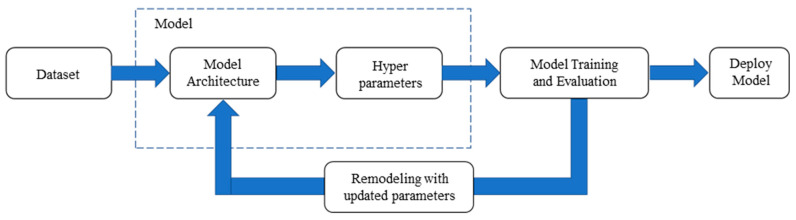
Training steps involved in ML.

**Figure 5 sensors-22-09783-f005:**
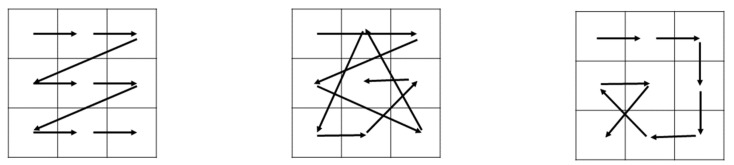
Grid search (**left**), random search (**middle**), and Bayesian search (**right**).

**Figure 6 sensors-22-09783-f006:**
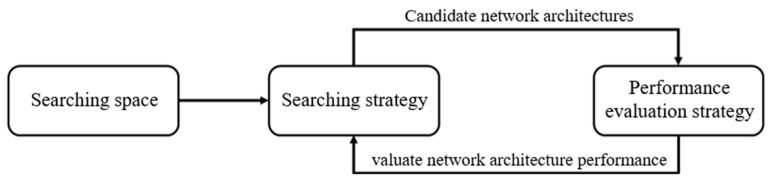
Three components of Google’s NAS system.

**Figure 7 sensors-22-09783-f007:**
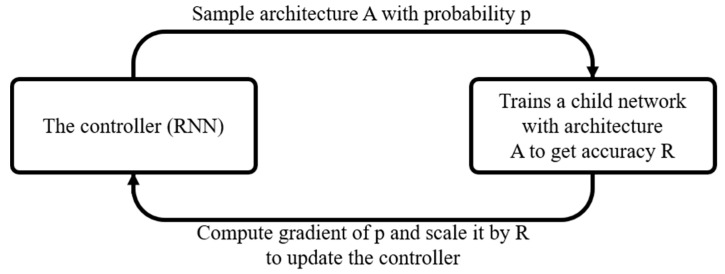
NASNet operation [25].

**Figure 8 sensors-22-09783-f008:**
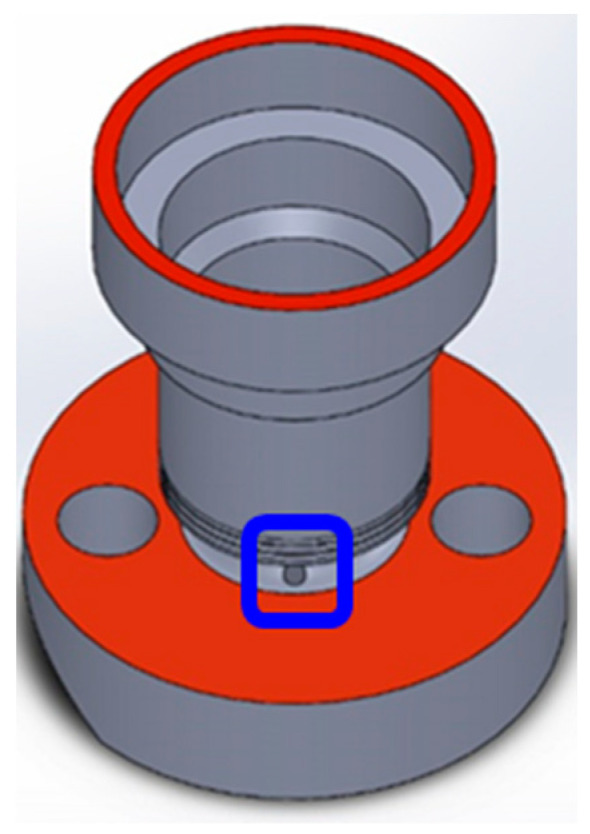
Workpiece and detection target (circular hole) indicated by the blue square.

**Figure 9 sensors-22-09783-f009:**
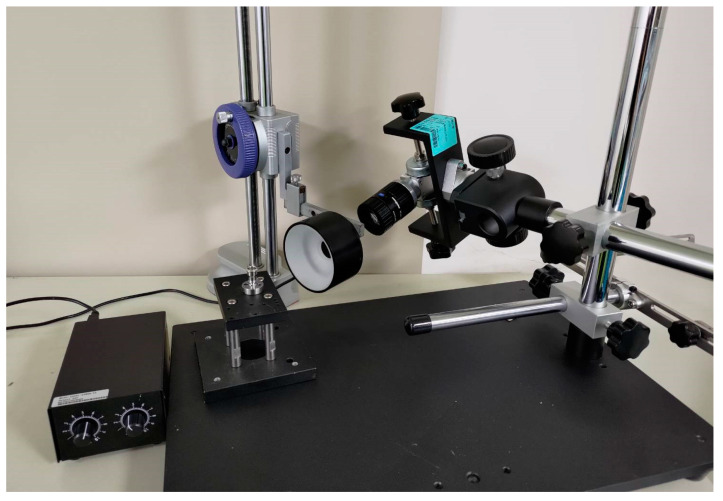
Illustration of the camera and light source positioning.

**Figure 10 sensors-22-09783-f010:**
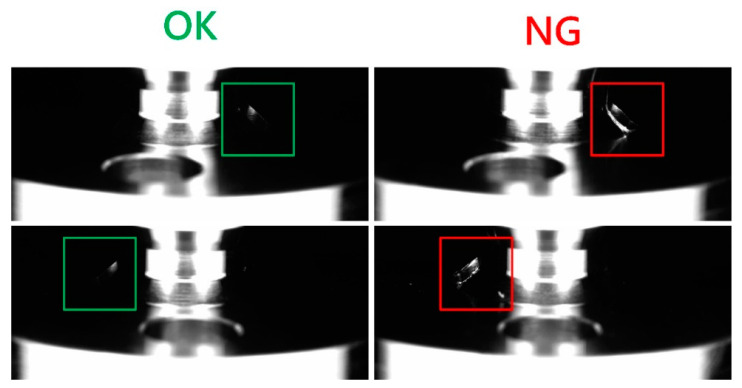
Original OK (**left**) and NG defect (**right**) image.

**Figure 11 sensors-22-09783-f011:**
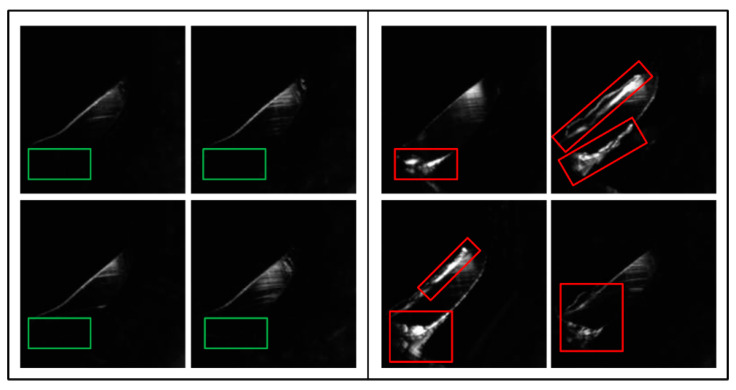
Cropped OK image (**left**) and NG defect (**right**) training images.

**Figure 12 sensors-22-09783-f012:**
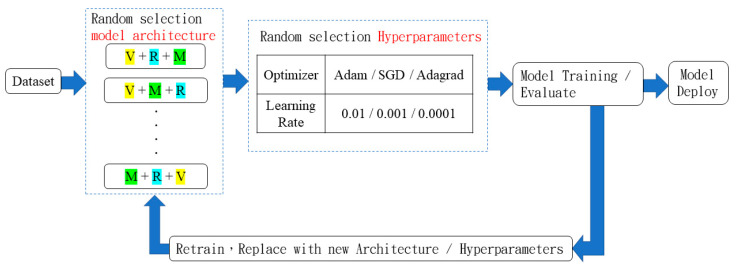
Operation of designed AutoML model.

**Figure 13 sensors-22-09783-f013:**
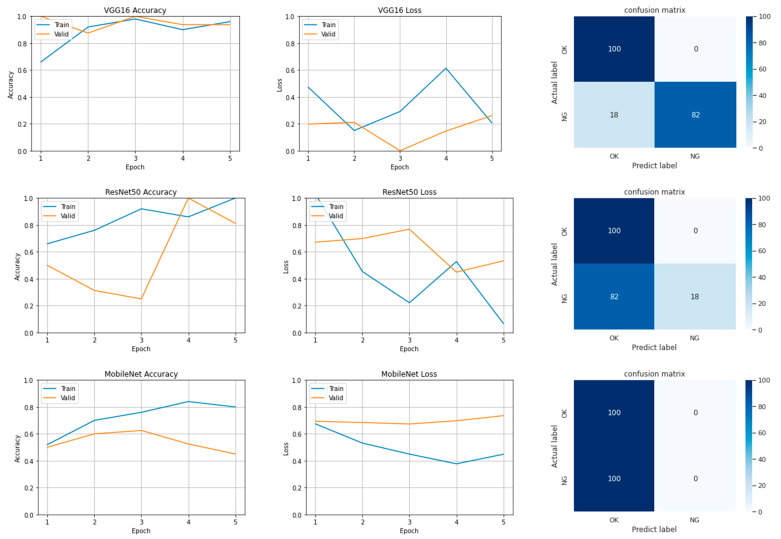
Accuracy and loss in training and validation and confusion matrix for each tested model.

**Figure 14 sensors-22-09783-f014:**
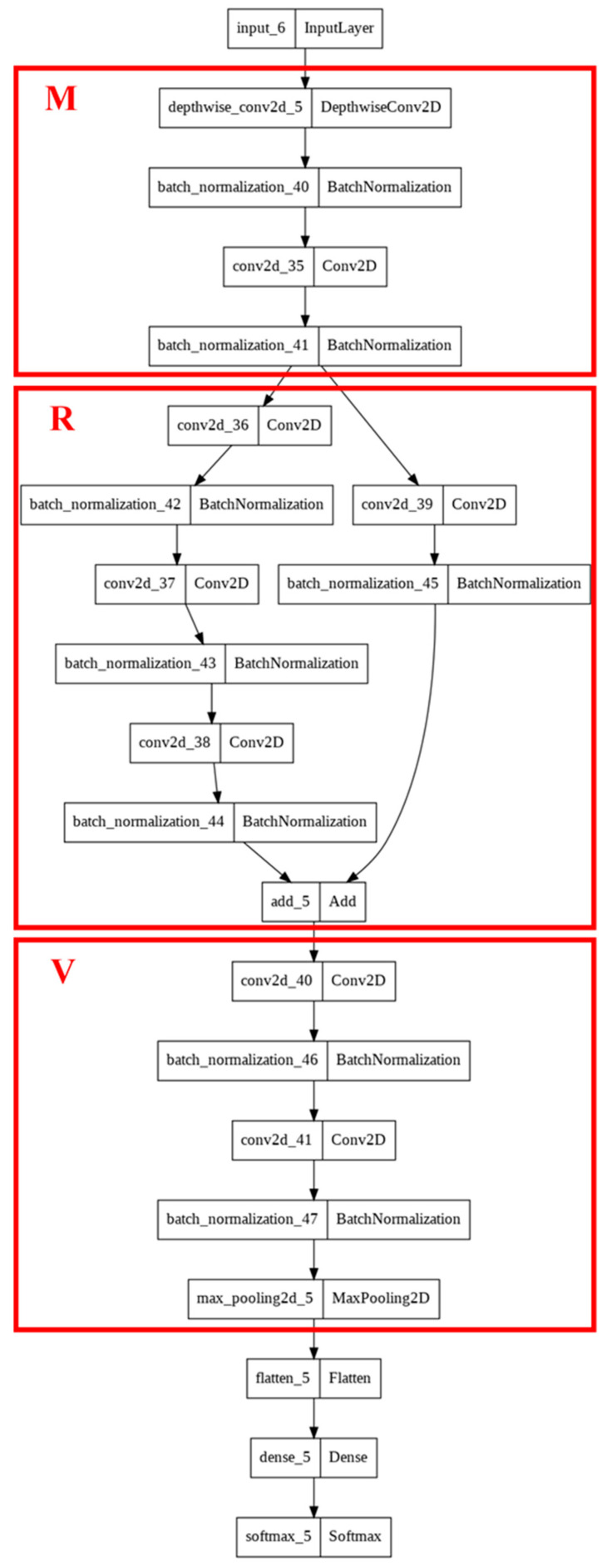
Structure of the designed AutoML model based on MRV.

**Figure 15 sensors-22-09783-f015:**
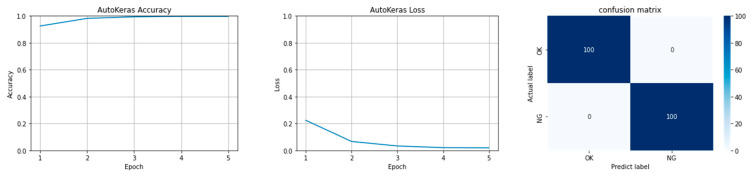
Accuracy, loss, and confusion matrix for AutoKeras model.

**Table 1 sensors-22-09783-t001:** Architecture of VGG-16 [18].

Layer Type	Filters	Kernel Size/Stride	Output Size	VGG Block
Conv 1 + Relu	64	3 × 3/1	224 × 224 × 64	1
Conv 2 + Relu	64	3 × 3/1	224 × 224 × 64	
MaxPooling			112 × 112 × 64	
Conv 3 + Relu	128	3 × 3/1	112 × 112 × 128	2
Conv 4 + Relu	128	3 × 3/1	112 × 112 × 128	
MaxPooling			56 × 56 × 128	
Conv 5 + Relu	256	3 × 3/1	56 × 56× 256	3
Conv 6 + Relu	256	3 × 3/1	56 × 56× 256	
Conv 7 + Relu	256	3 × 3/1	56 × 56× 256	
MaxPooling			28 × 28× 256	
Conv 8 + Relu	512	3 × 3/1	28 × 28 × 512	4
Conv 9 + Relu	512	3 × 3/1	28 × 28 × 512	
Conv 10 + Relu	512	3 × 3/1	28 × 28 × 512	
MaxPooling			14 × 14 × 512	
Conv 11 + Relu	512	3 × 3/1	14 × 14 × 512	5
Conv 12 + Relu	512	3 × 3/1	14 × 14 × 512	
Conv 13 + Relu	512	3 × 3/1	14 × 14 × 512	
MaxPooling			7 × 7 × 512	
Fully Connected 1			1 × 1 × 4096	
Fully Connected 2			1 × 1 × 4096	
Fully Connected 3			1 × 1 × 1000	
Softmax				

**Table 2 sensors-22-09783-t002:** Architecture of ResNet [12].

layer name	output size	18-layer	34-layer	50-layer	101-layer	152-layer
conv1	112×112	7×7, 64, stride 2				
conv2_x	56×56	3×3 max pool, stride 2				
		$\left[\begin{array}{cc} 3\times 3 & 64\\ 3\times 3 & 64 \end{array}\right]\times 2$	$\left[\begin{array}{cc} 3\times 3 & 64\\ 3\times 3 & 64 \end{array}\right]\times 3$	$\left[\begin{array}{cc} 1\times 1 & 64\\ 3\times 3 & 64\\ 1\times 1 & 256 \end{array}\right]\times 3$	$\left[\begin{array}{cc} 1\times 1 & 64\\ 3\times 3 & 64\\ 1\times 1 & 256 \end{array}\right]\times 3$	$\left[\begin{array}{cc} 1\times 1 & 64\\ 3\times 3 & 64\\ 1\times 1 & 256 \end{array}\right]\times 3$
conv3_x	28×28	$\left[\begin{array}{cc} 3\times 3 & 128\\ 3\times 3 & 128 \end{array}\right]\times 2$	$\left[\begin{array}{cc} 3\times 3 & 128\\ 3\times 3 & 128 \end{array}\right]\times 4$	$\left[\begin{array}{cc} 1\times 1 & 128\\ 3\times 3 & 128\\ 1\times 1 & 512 \end{array}\right]\times 4$	$\left[\begin{array}{cc} 1\times 1 & 128\\ 3\times 3 & 128\\ 1\times 1 & 512 \end{array}\right]\times 4$	$\left[\begin{array}{cc} 1\times 1 & 128\\ 3\times 3 & 128\\ 1\times 1 & 512 \end{array}\right]\times 8$
conv4_x	14×14	$\left[\begin{array}{cc} 3\times 3 & 256\\ 3\times 3 & 256 \end{array}\right]\times 2$	$\left[\begin{array}{cc} 3\times 3 & 256\\ 3\times 3 & 256 \end{array}\right]\times 6$	$\left[\begin{array}{cc} 1\times 1 & 256\\ 3\times 3 & 256\\ 1\times 1 & 1024 \end{array}\right]\times 6$	$\left[\begin{array}{cc} 1\times 1 & 256\\ 3\times 3 & 256\\ 1\times 1 & 1024 \end{array}\right]\times 23$	$\left[\begin{array}{cc} 1\times 1 & 256\\ 3\times 3 & 256\\ 1\times 1 & 1024 \end{array}\right]\times 36$
conv5_x	7×7	$\left[\begin{array}{cc} 3\times 3 & 512\\ 3\times 3 & 512 \end{array}\right]\times 2$	$\left[\begin{array}{cc} 3\times 3 & 512\\ 3\times 3 & 512 \end{array}\right]\times 3$	$\left[\begin{array}{cc} 1\times 1 & 512\\ 3\times 3 & 512\\ 1\times 1 & 2048 \end{array}\right]\times 3$	$\left[\begin{array}{cc} 1\times 1 & 512\\ 3\times 3 & 512\\ 1\times 1 & 2048 \end{array}\right]\times 3$	$\left[\begin{array}{cc} 1\times 1 & 512\\ 3\times 3 & 512\\ 1\times 1 & 2048 \end{array}\right]\times 3$
	1×1	average pool, 1000-d fc, softmax				
FLOPs		$1.8\;\times \;10^{9}$	$3.6\;\times \;10^{9}$	$3.8\;\times \;10^{9}$	$7.6\;\times \;10^{9}$	$11.3\;\times \;10^{9}$

**Table 3 sensors-22-09783-t003:** Architecture of MobileNet v1 [13].

Type/Stride		Filter Shape	Input Size
Conv/s2		3 × 3 × 3 × 32	224 × 224 × 3
Conv dw/s1		3 × 3 × 32 dw	112 × 112 × 32
Conv/s1		1 × 1 × 32 × 64	112 × 112 × 32
Conv dw/s2		3 × 3 × 64 dw	112 × 112 × 64
Conv/s1		1 × 1 × 64 × 128	56 × 56 × 64
Conv dw/s1		3 × 3 × 128 dw	56 × 56 × 128
Conv/s1		1 × 1 × 128 × 128	56 × 56 × 128
Conv dw/s2		3 × 3 × 128 dw	56 × 56 × 128
Conv/s1		1 × 1 × 128 × 256	28 × 28 × 128
Conv dw/s1		3 × 3 × 128 dw	28 × 28 × 256
Conv/s1		1 × 1 × 256 × 256	28 × 28 × 256
Conv dw/s2		3 × 3 × 256 dw	28 × 28 × 256
Conv/s1		1 × 1 × 256 × 512	14 × 14 × 256
5 ×	Conv dw/s1	3 × 3 × 512 dw	14 × 14 × 512
	Conv/s1	1 × 1 × 512 × 512	14 × 14 × 512
Conv dw/s2		3 × 3 × 512 dw	14 × 14 × 512
Conv/s1		1 × 1 × 512 × 1024	7 × 7 × 512
Conv dw/s2		3 × 3 × 1024 dw	7 × 7 × 1024
Conv/s1		1 × 1 × 1024 × 1024	7 × 7 × 1024
Avg pool/s1		Pool 7 × 7	7 × 7 × 1024
FC/s1		1024 × 1000	1 × 1 × 1024
Softmax/s1		Classifier	1 × 1 × 1000

**Table 4 sensors-22-09783-t004:** Results of Experiment 1.

	VGG-16	ResNet-50	MobileNet v1
Accuracy	0.9100	0.5900	0.5000
Loss	0.2280	0.6290	0.7144
Number of parameters	134,268,738	23,591,810	3,230,914
Training time (s)	538	195	77

**Table 5 sensors-22-09783-t005:** Training results obtained ith VGG-16, ResNet-50, and MobileNet v1 with two data distributions.

	VGG-16		ResNet-50		MobileNet v1	
Distribution ratio	five-five	eight-two	five-five	eight-two	five-five	eight-two
Accuracy (%)	91.00	66.39	59.00	56.94	50.00	44.44
Loss (%)	22.80	63.16	62.90	55.82	71.44	69.57
Time (s)	538	555	195	201	77	71
TP	100	20	100	20	100	0
TN	0	0	0	0	0	20
FP	18	20	82	20	100	0
FN	82	0	18	0	0	20

**Table 6 sensors-22-09783-t006:** Parameters and architectures selected by designed AutoML model in six training iterations.

	First Time	Second Time	Third Time	Fourth Time	Fifth Time	Sixth Time
Learning rate	0.0001	0.0001	0.0001	0.01	0.0001	0.001
Optimizer	Adam	Adagrad	Adagrad	SGD	SGD	Adam
Model composition order	RMV	RMV	MVR	VMR	RMV	MRV
Number of parameters	928,450	928,450	2,019,116	1,901,378	928,450	810,796
Training time (s)	1911	1897	4171	1013	1917	2339
Accuracy	0.6600	0.7400	0.9000	0.8750	0.8250	0.9550
Loss	0.6399	0.5238	1.1061	0.3181	0.2680	0.5756
TP	100	100	100	100	100	96
TN	0	0	0	0	0	4
FP	68	52	20	25	35	5
FN	32	48	80	75	65	95

**Table 7 sensors-22-09783-t007:** Model architecture selected in sixth training iteration by designed AutoML model.

Layer Type	Filter Shape	Output Size
Conv_dw_BN_ReLU	3×3×512 dw, Stride 2	224×224×3
Conv_BN_ReLU	1×1×512	224×224×512
Conv 3_BN_ReLU	$\left[\begin{array}{cc} 1\times 1 & 64\\ 3\times 3 & 64\\ 1\times 1 & 256 \end{array}\right]\times 1$	112×112×512
Conv 1_ReLU	3×3, 64	112×112×64
Conv 2_ReLU	3×3, 64	112×112×64
MaxPooling	2×2, Stride 2	56×56×64
Flatten		200,704
Fully Connected		2
Softmax		2

**Table 8 sensors-22-09783-t008:** Model architecture selected by AutoKeras.

Layer Type	Filter Shape	Output Size
Normalization		256 × 256 × 3
Conv1	3 × 3 × 32	254 × 254 × 32
Conv2	3 × 3 × 64	252 × 252 × 64
MaxPooling	2 × 2, Stride 2	126 × 126 × 64
Dropout (0.25)		126 × 126 × 64
Flatten		1,016,064
Dropout (0.5)		1,016,064
Dense		1

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
