# Peer review of "Automated Machine Learning System for Defect Detection on Cylindrical Metal Surfaces"

_sensors, 2022, doi:10.3390/s22249783_

Round 1

Reviewer 1 Report

I have severals questions:

1.- The images were cropped  to 186 × 189 pixels. Have other sizes been tested?

2.- There are many true negatives (TN) in the classification results. 

3.- Other parameters should be included to show the goodness and robustness of the system.

4.- Has the algorithm been tested on any subset of data, completely independent of the training and validation data?

5.- Bibliographical references should be updated

Author Response

Reviewer # 1

I have several questions:

Response:

The authors fully appreciate this reviewer's comments and suggestions. Authors’ response to esteemed reviewers’ questions is as followed.

Concern 1:

The images were cropped to 186 × 189 pixels. Have other sizes been tested?

Response:

The authors appreciate this reviewer's suggestions. The reason for resizing the image data to 186×189 pixels is to meet the needs for model calculations. Indeed, cropping to other pixel sizes can also be a selected choice. In this study, image sizes (186x189) for all different test models were the same. We will consider this precious suggestion as bettering our CNN parameters for future work.

Concern 2:

There are many true negatives (TN) in the classification results.

Response:

The authors agree with the reviewer’s comment. Results of many true negatives (TN) in the classification results are true. A true negative is the outcome result where the model correctly predicts the true negative class. In short, the higher the TN rate, the higher the accuracy of our designed AutoML model. (lines 305-306, revised paper: TPs were OK images correctly identified as OK. TNs were NG images correctly identified as No Good.)

Concern 3:

Other parameters should be included to show the goodness and robustness of the system.

Response:

The authors fully appreciate this reviewer's suggestions. In this study, the selection of hyperparameters of our designed AutoML was refer to the AutoKeras heuristically. The AutoKeras selects the optimizer and learning rate only as their parameters. In the future, other hyperparameter options such as batch size, epoch etc. can be included as an overall evaluation to show the goodness and robustness of our designed AutoML system. We thank again for reviewer's suggestions.

Concern 4:

Has the algorithm been tested on any subset of data, completely independent of the training and validation data?

Response:

The authors fully appreciate this reviewer's comments and suggestions. Yes, the tested data were subset of data which were independent of the training and validation data.

Concern 5:

Bibliographical references should be updated

Response:

The authors fully appreciate this reviewer's suggestions and we did update two bibliographical references in revised paper. (Line 53, with red).

Reviewer 2 Report

This manuscript reported an automated machine learning network for defect detection. Retraining criterion is developed to select and optimize hyperparameters for a new task. And AutoKeras was used to execute AutoML and identify a suitable model for defect dataset. Finally, defective and low-quality samples can be distinguished with a low training cost. Comparative experimental results demonstrated the feasibility of the proposed method. However, I think the structure of this manuscript can be improved and more kinds of defects are suggested to be tested and shown. And my specific comments are as follows.

1.       Section 2 “Introduction to VGGNet, ResNet, MobileNet, and AutoML” is too long and detailed, but all of these contents are previous existing works. So, this part is suggested to be simplified or posted as supplements.

2.       Compared with the Section 2, Section 3.4. “Design of AutoML Model” is the new materials, but description of this part is too simple. More details are suggested to be added, such as how to select and determine the model architecture, optimizer, and learning rate, how to update new hyperparameters and what is the termination criteria of the retrain.

3.       More kinds of defects should be shown in dataset and the actual tested defects are also suggested to be shown to demonstrate the transfer ability of the proposed method.

4.       Direct comparisons and improvement on training cost and transfer ability of the proposed AutoML model compared with the traditional VGG-16, ResNet-50, and MobileNet v1 model is suggested to be given.  

5.       There is a missing part about defects in Fig. 8.

Author Response

Reviewer # 2

This manuscript reported an automated machine learning network for defect detection. Retraining criterion is developed to select and optimize hyperparameters for a new task. And AutoKeras was used to execute AutoML and identify a suitable model for defect dataset. Finally, defective and low-quality samples can be distinguished with a low training cost. Comparative experimental results demonstrated the feasibility of the proposed method. However, I think the structure of this manuscript can be improved and more kinds of defects are suggested to be tested and shown. And my specific comments are as follows.

Response:

The authors fully appreciate this reviewer's comments and suggestions. Authors’ response to reviewers’ questions is as followed.

Concern 1:

Section 2 “Introduction to VGGNet, ResNet, MobileNet, and AutoML” is too long and detailed, but all of these contents are previous existing works. So, this part is suggested to be simplified or posted as supplements.

Response:

The authors fully appreciate this reviewer's suggestions. We did simplify some of Section 2 in the revised paper. In this manuscript, the design AutoML is based on the known knowledge and selected key module from the existing VGGNet, ResNet, MobileNet models. Therefore, we put more effort in section 2 and do the experimental results in Section 4.2. As an example, in Figure 14 we detail the structure of the designed AutoML model made by MRV (key neural network core layers from MobileNet, ResNet and VGGNet).

Concern 2:

Compared with the Section 2, Section 3.4. “Design of AutoML Model” is the new materials, but description of this part is too simple. More details are suggested to be added, such as how to select and determine the model architecture, optimizer, and learning rate, how to update new hyperparameters and what is the termination criteria of the retrain.

Response:

The authors fully appreciate this reviewer's comments and suggestions.

 These three VGG-16, ResNet-50, and MobileNet v1models were hypothetically implemented as for transfer learning (TL) first. However, deploying the TL, we found the phenomenon of apparent convergence in prediction accuracy followed by divergence in validation accuracy did create a problem when the image pattern is not known in advance. (lines 18-20, in abstract section)

That is, if we use any one of the architecture from the VGG-16, ResNet-50, and MobileNet v1, the prediction results could not be satisfied. And most likely it will not converge even though we did a lot of efforts for the hyperparameters tuning. Therefore, our developed automated machine learning (AutoML) model adopts the core layers network architecture from the three VGG-16, ResNet-50, and MobileNet v1 models. Each core layer contributes its significant feature with its known knowledge.  

Most often, the topology of neural network NN is unknown. Thus, in this manuscript, our designed AutoML is invented. Since this designed NN structure was manipulated and based on selected key module from the existing VGGNet, ResNet, MobileNet models. Therefore, we have to put more efforts in section 2, and give an overall picture of our design concept in Section 3.4. But, we go more details in the experimental results from Section 4.1 and then go to Section 4.2 where we delivered the step-by-step for this new method. Then, the reader can get a solid solution by how to build a NN structure in Section 4.2.  As an example, in Figure 12 the operation of designed AutoML model is depicted.  Figure 14 we detail and show the NN structure of the designed AutoML model made by MRV.

As for the issues of how the select the optimizer, learning rate, how to update new hyperparameters, the termination criteria of the retrain were not discussed in this study. In Figure 12, the hyperparameters for the operation of designed AutoML model is selected randomly. The authors fully agree this reviewer's comments and suggestions. We will use genetic algorithm, particle swarm optimization or Taguchi method as a tool and move on this precious issues as future works.

Concern 3:

More kinds of defects should be shown in dataset and the actual tested defects are also suggested to be shown to demonstrate the transfer ability of the proposed method.

Response:

The authors fully appreciate this reviewer's suggestions and regret that we failed to do our best on this study. This dataset and the interested defect images were provided by industry company. Increasing the defect classifications is a need. Indeed, it is one of the future goals of our research. But unfortunately, we don't have another kind of defect dataset for demonstration at present.  We thank again for this valuable suggestion.

Concern 4:

Direct comparisons and improvement on training cost and transfer ability of the proposed AutoML model compared with the traditional VGG-16, ResNet-50, and MobileNet v1 model is suggested to be given.

Response:

The authors regret that we did not explain our work well.

First, we demonstrated that the TL for image defect detection by using VGG-16, ResNet-50, and MobileNet v1 models could not be satisfied within five epochs. Since deploying the TL, the phenomenon of apparent convergence in prediction accuracy followed by divergence in validation accuracy did create a problem when the image pattern is not known in advance.

Secondly, since the prediction accuracy is the main concern for defect detection in this study, in Table 4 and Table 5, the highest prediction accuracy 91% by use of VGG-16 is set as the goal for our designed AutoML.  Even though the number of parameters and training time of VGG-16 is the highest.  

Thirdly, based on the aim of prediction accuracy must higher than the 91%, we abstract the core module layers developed by each VGG-16, ResNet-50, and MobileNet v1.  

In Table 6, we illustrated that with the operation of our designed AutoML model in Figure 12, the self-designed AutoML model has bettering accuracy by 0.955 than 0.91 via VGG-16. Compare Table 4 and Table 6, the number of AutoML’s parameters compared with the VGG-19 model was 810,796 vs. 134,268,738 although the training time was 2339 vs.538 seconds. Both of VGG-19 and our AutoML used the same Adam optimizer, batch size of 10, number of epochs by 5, and same 0.0001 learning rate.

Concern 5:

There is a missing part about defects in Fig. 8

Response:

The authors fully appreciate this reviewer's suggestions and we did replot it in the revised paper.

Round 2

Reviewer 2 Report

Authors' response addressed my concerns. And thus I recommend to accept the current version.